

# Awareness and attitudes towards eye donation among medical and allied health students in Jeddah, Saudi Arabia

Mahmood Showail[1,2,*], Turki A. AlAmoudi[2,*], Esraa Basalem[2], Khalid Alshebl[2], Nawaf Meshal Almalki[2], Abdullah Al Matrafi[2] and Mohammed Ashour[2]

[1] Department of Ophthalmology, King Abdul Aziz University, Jeddah, Saudi Arabia
[2] Faculty of Medicine, King Abdul Aziz University, Jeddah, Saudi Arabia
* These authors contributed equally to this work.

## ABSTRACT

**Background:** Many national studies in Saudi Arabia have revealed a lack of knowledge about eye donation. The current study assessed awareness and attitudes towards eye donation among health faculty students in Jeddah, Saudi Arabia. It aims to increase their awareness as future healthcare providers are expected to raise general awareness to attain more local corneal donations.

**Methods:** A cross-sectional study including 1,060 health faculty students was conducted at King Abdulaziz University. Data were collected through an online questionnaire that covered participants' demographics, academic year, knowledge, and attitudes regarding eye donation.

**Results:** Thirty-five percent of students had heard about eye donation, with the most common sources of information being social media (29%) and health workers (24%). Most respondents, 61% ($n = 643$), indicated willingness to donate their eyes after death. Of these respondents, 93% ($n = 986$), 6.8% ($n = 72$), and 0.2% ($n = 2$) had poor, fair, and good knowledge levels, respectively. A total of 66% ($n = 696$) and 34% ($n = 364$) had negative and positive attitudes, respectively.

**Conclusion:** Students in this study showed low levels of knowledge and negative attitudes about eye donation. Students should be adequately educated about the significance of eye donation.

## INTRODUCTION

The eye's main structural and immunological barrier is the cornea, a transparent, avascular connective tissue. It also serves as the anterior refractive surface of the eye, along with the tear film that lies overtop (*DelMonte & Kim, 2011*). Damage to the cornea due to disease or injury can lead to poor vision or blindness. Keratoplasty, or corneal transplantation, is a surgical procedure for corneal blindness that restores vision; during this operation, the damaged cornea is replaced by a donor cornea (*Magdum et al., 2014*).

Corresponding author
Turki A. AlAmoudi,
turkiasalamoudi@gmail.com

The International Agency for the Prevention of Blindness (IAPB) reported an estimated 43 million people with blindness around the world in 2020; besides cataract (39.5%), refractive errors (8.5%), glaucoma (8.3%), macular degeneration (4.3%), and diabetic retinopathy (2.4%) there 16 million (37%) were listed with 'residual causes for blindness' where corneal diseases came as one of the leading causes of blindness among all ages around the world (*GBD 2019 Blindness and Vision Impairment Collaborators, Vision Loss Expert Group of the Global Burden of Disease Study, 2021*; *Flaxman et al., 2017*). Corneal scarring is the primary factor causing reversible blindness in children worldwide, and it is one of the leading causes of unilateral and bilateral corneal blindness (*Tabbara & Ross-Degnan, 1986*; *Tabbara, 2001*).

Corneal transplant is the primary approach for helping people with corneal blindness regain their vision, and there are many different disorders for which donor corneas are indicated (*Oliva, Schottman & Gulati, 2012*). Corneas are the most commonly transplanted tissues; according to the Eye Bank Association of America (EBAA), 49,597 corneal transplants were performed in 2022 (*Eye Bank Association of America, 2022*; *Tan et al., 2012*). More than 90% of corneal transplant procedures are successful and sight-saving. Still, the number of surgeries that can be performed is constrained by the lack of corneal tissue donors, a severe problem in developing countries (*Tan et al., 2012*; *Waziri-Erameh, Ernest & Edema, 2007*). Raising public awareness can help remove obstacles and boost the number of eye donors (*Gupta et al., 2017*; *Krishnaiah et al., 2004*; *Tandon et al., 2004*; *Yew et al., 2005*).

The corneal tissue donation and transplantation program was established in Saudi Arabia in 1983. Since then, until 2020, only 715 corneas were locally recovered from deceased organ donors, where corneal recovery was usually made along with a multi-organ retrieval procedure from brainstem death donors. It is worth mentioning that there are 10 corneal transplant centers, including three centers with corneal banks in Saudi Arabia; nevertheless, most corneas are imported from abroad to meet the local demand as in 2019 only, there were 1,895 corneas imported mainly from the United States (*Saudi Center for Organ Transplantation, 2020*).

Many national studies conducted in Saudi Arabia revealed a low level of public awareness about eye donation, which impacts the number of donors. Hence, the purpose of this study was to evaluate awareness and attitudes towards the practice of eye donation among students of the health faculties' in Jeddah, Saudi Arabia, to increase their awareness level as future healthcare providers who are expected to raise public awareness to achieve more local corneal donations.

## MATERIALS AND METHODS

**Study design and setting:** A cross-sectional study was conducted at King Abdulaziz University in Jeddah, Saudi Arabia, in May 2022. All data were obtained by online questionnaires in Google Forms, which were distributed to participants after they received a complete explanation of the aim of this study, and informed consent was obtained before filling it out. To be included in the study, students had to be undergraduates in one of the following departments: medicine, dentistry, pharmacy, nursing, applied medical sciences,

or medical rehabilitation. Students were excluded from the study if they were undergraduates in other departments, interns, postgraduates, or enrolled outside King Abdulaziz University.

**Sampling techniques:** We randomly invited 1,120 health faculty students, but 1,060 consented and completed the survey. The sample size was calculated using a sample size calculator at Calculator.net (https://www.calculator.net/sample-size-calculator.html). The confidence level was 95%, the margin of error was 5%, and the population size was 10,000, assuming that 50% of participants had good knowledge and a positive attitude towards eye donation. The calculated sample size was 370, but considering students' non-response rates, more students were invited to participate.

**Questionnaire and data collection:** A pre-designed online questionnaire was used to collect data. The first part included items about the participants' demographics and academic year. The second part included 11 items to assess the participants' knowledge and four to determine their attitude about eye donation. The participant was classified as having a poor knowledge level if scored <50% of the correct answers for knowledge items, having fair knowledge if scored 50–75%, and having good knowledge if scored >75%. The same scoring was followed for the attitude scoring (*Alzuhairy et al., 2020*).

**Validity and reliability analysis:** Questions in the study questionnaire covered the level of knowledge and attitude towards eye donation. The questionnaire was adapted from previous studies (*Alzuhairy et al., 2020*; *Taber, 2018*). Twenty participants who were not included in the final study results were involved in a pilot study, and the results were utilized to analyze reliability and validity.

**Questionnaire validation:** A panel of three experts evaluated the questionnaire's validity. To determine whether the original items were appropriate for examination. The objective assigned to the subject matter experts was to assess each item's applicability and relevance on a four-point scale: 1 = not so adequate (can be omitted); 2 = needs major modification; 3 = adequate but needs minor revision; and 4 = adequate (simple, relevant, and clear). The content validity index (CVI) is the proportion of all items with a score of 3 or 4 from experts. A score is considered to have good validity 80% of the time. The CVI of the planned questionnaire was determined.

**Reliability analysis:** When internal consistency reliability was investigated, a Cronbach's alpha value of 0.82 was found. When a scale's Cronbach's alpha value is greater than 0.83, the scale is regarded as being internally consistent (*Dandona et al., 1999*).

**Data analysis:** Statistical Package for the Social Sciences Software (SPSS) version 26 (IBM Corp. Armonk, NY, USA) was used to assess the relationship between variables. Qualitative data was expressed as numbers and percentages. The chi-square test ($\chi^2$) was used for data analysis. Mean and standard deviation (mean ± SD) were used to express the quantitative variables. Mann-Whitney and Kruskal Wallis tests were used to test nonparametric variables. Correlation analysis was performed using Spearman's test, and a *p*-value < 0.05 was considered statistically significant.

**Confidentiality and ethical approval:** The study was reviewed and approved by the Unit of Biomedical Ethics, Research Ethics Committee (REC), KAUH, Jeddah, Saudi Arabia (Approval No.: 14440624, Reference No.: 612-22). Identifying information such as

participant names, contact numbers, and addresses was not asked in the questionnaire to guarantee the confidentiality of the participants.

## RESULTS

This cross-sectional study was conducted among health faculty students at King Abdulaziz University. A total of 1,060/1,120 participated in this study, with a response rate of 94%. The mean age of participants was 21.47 ± 1.68 years; 76% ($n = 807$) were male, and 98.5% ($n = 1,044$) had a Saudi nationality. The majority were from the faculty of medicine, 42% ($n = 446$), and 24% ($n = 250$) were in their sixth academic year (Table 1).

Only 35% ($n = 371$) of participants had previously heard about eye donation. Of these, 21% ($n = 222$) knew that the eye parts used were the cornea and sclera. About 46% (n = 492) correctly knew that a live donor could not make eye donation, and 41% ($n = 439$) knew that the ideal time for donating eyes is 0–12 h after death, 34% ($n = 360$) correctly knew that it has no age limit, and 17% ($n = 181$) knew that contagious diseases are a contraindication for eye donation. About 40% ($n = 429$) correctly knew that blood group matching between a donor and a recipient was unnecessary. Only 6% ($n = 59$) knew that the donor eye could be preserved/stored for 12 days before transplantation, and 32% ($n = 339$) correctly knew that donors provided donation consent while alive or by their relatives after their death. In contrast, only 9% ($n = 95$) were aware of any eye bank in Saudi Arabia, 19% ($n = 202$) knew where to apply if they wanted to register for eye donation, and only 3% ($n = 33$) thought that they have enough information regarding eye donation (Table 2).

The most common sources of information about eye donation were social media, 29% ($n = 304$), followed by health workers, 24% ($n = 259$) (Table 3). The most preferred sources for receiving more information were eye donation campaign 92% ($n = 973$) and social media 81% ($n = 860$) (Table 4).

About the participants' attitudes, the majority, 61% ($n = 643$), indicated that they were willing to donate their eyes after death. For those unwilling or did not know, 11% ($n = 115$) would change their decision if allowed to donate corneas only. Almost 45% ($n = 476$) were willing to donate the eyes of their family members (first-degree relatives) after death. Most of them, 74% ($n = 785$), would educate others about eye donation if they had more information (Table 5).

As for the motives behind the donation, most of the participants, 88% ($n = 936$), were willing to donate to do a good act and obtain a religious reward. In addition, 72.5% ($n = 769$) were willing to serve scientific research and advancement. On the other hand, 15% ($n = 157$) believed that eye donation is forbidden in their religion, while 16% ($n = 168$) believed it is culturally unacceptable, and 22% ($n = 230$) thought it might disfigure the donor's face. Most of them, 69% ($n = 733$), agreed that they lacked knowledge about eye donation, and 65.5% ($n = 694$) did not know how and where to apply for donation (Table 6).

The mean knowledge score was 3 ± 1.66, and the mean attitude score was 1.9 ± 0.98. As for knowledge level, 93% (n = 986), 6.8% ($n = 72$), and 0.2% ($n = 2$) had poor, fair, and

**Table 1 Demographic characteristics of the participants.**

| Variable | Mean ± SD, or $n$ (%) |
|---|---|
| Age, years | 21.47 ± 1.68 |
| **Gender** | |
| Male | 807 (76.1) |
| Female | 253 (23.9) |
| **Nationality** | |
| Saudi | 1,044 (98.5) |
| Non-Saudi | 16 (1.5) |
| **Faculty** | |
| Medicine | 446 (42.1) |
| Dentistry | 218 (20.6) |
| Pharmacy | 183 (17.3) |
| Nursing | 130 (12.3) |
| Applied medical sciences | 32 (3) |
| Medical rehabilitation (PT, OT, and RT) | 51 (4.8) |
| **Academic year** | |
| Second | 330 (31.1) |
| Third | 197 (18.6) |
| Fourth | 148 (14) |
| Fifth | 135 (12.7) |
| Sixth | 250 (23.6) |

Note:
Physiotherapy (PT). Occupational therapy (OT). Respiratory therapy (RT). The first academic year is a preparatory or pre-college year in the education system at KAU.

good knowledge levels, respectively. At the same time, 66% ($n = 696$) and 34% ($n = 364$) had negative and positive attitudes, respectively (Fig. 1).

Male participants who had heard about eye donation and did not think they had enough information were significantly more likely to have a good knowledge level ($p \leq 0.05$). In addition, participants who had their information about eye donation from friends/relatives or social media also had a significantly higher likelihood of having a good knowledge level ($p \leq 0.05$) (Table 7).

Male medical students in the second academic year with Saudi nationality who did not obtain their information from a health worker were significantly more likely to have a positive attitude ($p \leq 0.05$) (Table 8). There was a significant relationship between knowledge level and attitude level ($p \leq 0.05$) (Fig. 2).

## DISCUSSION

Public attitude towards eye donation in Saudi Arabia faces challenges and obstacles similar to those faced by kidney donations a couple of decades ago (*Saudi Center for Organ Transplantation, 2020*), where misunderstanding and poor levels of knowledge are the leading obstacles.

**Table 2 Students knowledge.**

| Variable | n (%) |
|---|---|
| **Have you heard about eye donation?** | |
| **Yes** | **371 (35)** |
| No | 689 (65) |
| **Which eye parts are concerned with donation?** | |
| All eye parts | 451 (42.5) |
| **Cornea and sclera (correct answer)** | **222 (20.9)** |
| Optic nerve | 56 (5.3) |
| Retina | 71 (6.7) |
| I don't know | 260 (24.5) |
| **Can eye donation be done by a live donor?** | |
| Yes | 211 (19.9) |
| **No (correct answer)** | **492 (46.4)** |
| I don't know | 357 (33.7) |
| **When do you think is the ideal time for donating eyes after death?** | |
| **0–12 h (correct answer)** | **439 (41.4)** |
| 13–24 h | 178 (16.8) |
| 25–48 h | 122 (11.5) |
| I don't know | 321 (30.3) |
| **Is there any age limits for eye donation?** | |
| Yes | 336 (31.7) |
| **No (correct answer)** | **360 (34)** |
| I don't know | 364 (34.3) |
| **Which of the following is contraindication for eye donation?** | |
| Retinal diseases | 286 (27) |
| Cataract | 124 (11.7) |
| Glaucoma | 172 (16.2) |
| **Contagious diseases (correct answer)** | **181 (17.1)** |
| I don't know | 297 (28) |
| **Is it necessary to match the blood group (between donor and recipient)?** | |
| Yes | 280 (26.4) |
| **No (correct answer)** | **429 (40.5)** |
| I don't know | 351 (33.1) |
| **How long can the donor eye be preserved/stored before transplantation?** | |
| Less than 6 h | 272 (25.7) |
| Less than 24 h | 167 (15.8) |
| 7 days | 73 (6.9) |
| **12 days (correct answer)** | **59 (5.6)** |
| 1 month | 61 (5.8) |
| I don't know | 428 (40.4) |
| **Do you know who gives the donation consent?** | |
| Donors while alive | 485 (45.8) |
| Donors' relatives | 23 (2.2) |

| Variable | n (%) |
|---|---|
| Donors' friends | 5 (0.5) |
| **Donors while alive and their relatives (after death) (correct answer)** | **339 (32)** |
| I don't know | 208 (19.6) |
| **Are you aware of any eye bank in Saudi Arabia?** | |
| **Yes (correct answer)** | **95 (9)** |
| No | 965 (91) |
| **Do you know how and where to apply if you want to register for eye donation?** | |
| **Yes (correct answer)** | **202 (19.1)** |
| No | 858 (80.9) |
| **Do you think you have enough information regarding eye donation?** | |
| Yes | 33 (3.1) |
| No | 1027 (96.9) |

Note:
Participants' response to knowledge assessment questions ($N = 1,060$). Student responses worth 1 point on the scoring system that measures knowledge are in bold.

**Table 3 Sources of information.**

| Variable | n (%) |
|---|---|
| **What are the sources from which you obtained the information?** | |
| Health workers | 259 (24.4) |
| Friends/relatives | 192 (18.1) |
| Social media | 304 (28.7) |
| Television | 96 (9.1) |
| Eye donation campaign | 80 (7.5) |
| Posters | 84 (7.9) |

Note:
Distribution of the participants who heard about eye donation according to their sources of information ($N = 371$).

**Table 4 Preferable sources of information.**

| Variable | n (%) |
|---|---|
| **How do you prefer to receive more information regarding this topic?** | |
| Health workers | 747 (70.5) |
| Friends/relatives | 531 (50.1) |
| Social media | 860 (81.1) |
| Television | 742 (70) |
| Eye donation campaign | 973 (91.8) |
| Posters | 585 (55.2) |

Note:
Distribution of the participants according to their preferable sources of information ($N = 1,060$).

**Table 5 Students attitude.**

| Variable | n (%) |
|---|---|
| **Are you willing to donate your eyes after death?** | |
| Yes | **643 (60.7)** |
| No | 218 (20.6) |
| I don't know | 199 (18.8) |
| **(For those who chose I don't know or No) Regarding the last question, would you change your decision if you are given the option to donate your corneas only (leaving your eyes behind)? (N = 643)** | |
| Yes | **115 (10.8)** |
| No | 168 (12.6) |
| I don't know | 134 (60.7) |
| **Are you willing to donate your family members (first-degree relatives) eyes after death?** | |
| Yes | **476 (44.9)** |
| No | 283 (26.7) |
| I don't know | 301 (28.4) |
| **If you get more information, would you educate others about eye donation?** | |
| Yes | **785 (74.1)** |
| No | 55 (5.2) |
| Maybe | 220 (20.8) |

Note:
Participants' response to attitude assessment questions ($N = 1,060$). Student responses worth 1 point on the scoring system that measures attitude are in bold.

**Table 6 Reasons for donating and not donating eyes.**

| Variable | Strongly disagree | Disagree | Neutral | Agree | Strongly agree |
|---|---|---|---|---|---|
| **Possible reasons for donating eyes** | | | | | |
| To do good act and get religious reward | 13 (1.2) | 16 (1.5) | 95 (9) | 158 (14.9) | 778 (73.4) |
| To serve scientific research and development of science | 27 (2.5) | 55 (5.2) | 209 (19.7) | 168 (15.8) | 601 (56.7) |
| Know someone who is blind | 64 (6) | 82 (7.7) | 397 (37.5) | 130 (12.3) | 387 (36.5) |
| Know more information about eye donation | 22 (2.1) | 43 (4.1) | 428 (40.4) | 141 (13.3) | 426 (40.2) |
| **Possible reasons for not donating eyes** | | | | | |
| You believe it is forbidden in your religion | 268 (25.3) | 397 (37.5) | 238 (22.5) | 52 (4.9) | 105 (9.9) |
| You believe it is culturally unacceptable | 242 (22.8) | 361 (34.1) | 289 (27.3) | 107 (10.1) | 61 (5.8) |
| You think it might disfigure the donor's face | 175 (16.5) | 322 (30.4) | 333 (31.4) | 129 (12.2) | 101 (9.5) |
| Family objection | 123 (11.6) | 225 (21.2) | 348 (32.8) | 231 (21.8) | 133 (12.5) |
| Fear of unknown | 113 (10.7) | 224 (21.1) | 320 (30.2) | 251 (23.7) | 152 (14.3) |
| Lack of knowledge about eye donation | 52 (4.9) | 57 (5.4) | 218 (20.6) | 390 (36.8) | 343 (32.4) |
| Don't know how and where to apply for donation | 59 (5.6) | 65 (6.1) | 242 (22.8) | 372 (35.1) | 322 (30.4) |

Note:
Distribution of the participants according to possible reasons for donating and not donating eyes ($N = 1,060$).

Despite the limitations of this study, our findings are valuable in light of the growing demand for local corneal donors, where in the capital, Riyadh, as well as the eastern province of the kingdom, keratoconus was reported as the most common indication for corneal transplantation (*Al-Akily Saleh & Bamashmus Mahfouth, 2008*; *Alem et al., 2018*).

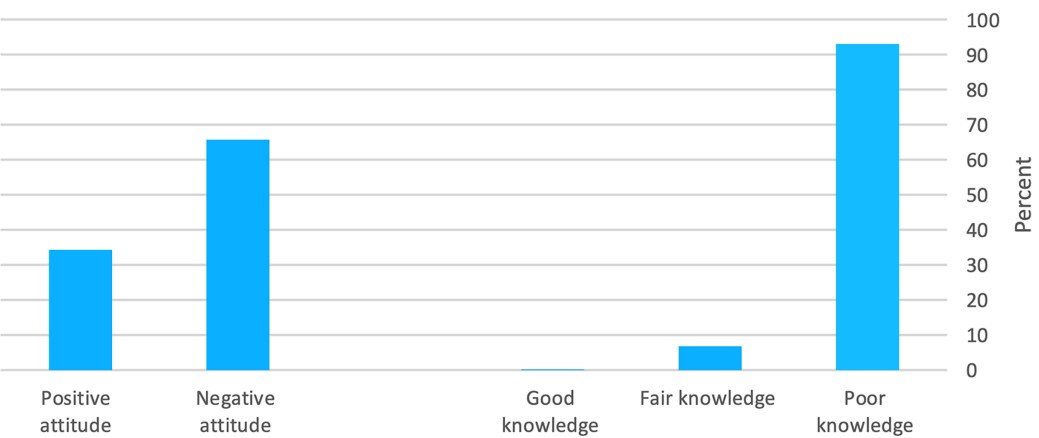

**Figure 1 Participants' percentage distribution according to their knowledge and attitude levels.**

This study found students' relatively low knowledge and awareness of eye donation. Only two (0.2%) had good knowledge, and the mean knowledge score was 3 ± 1.66 questions correct out of 11. This result was consistent with two studies among the general population in Saudi Arabia, 11 (2.9%) and 55 (4.3%) (*Alem et al., 2018*; *Alanazi et al., 2019*). However, this level of knowledge appears to be lower than that reported in another Saudi study including university students where 146 (35.8%) medical students at Taibah University were aware of eye donation (*Hameed & Jadidy, 2015*).

Another study revealed that 333 (97.9%) health sciences, medical, and nursing students in Goa, India, were aware of eye donation; this high level of awareness was attributed to the mass media and publicity campaigns, which were the main source of information (*Lal et al., 2018*).

A critically important issue that has to be highlighted to eye donors is the religious standpoint of organ donation. From this point of view, 668 (63%) participants knew that organ donation is not forbidden in many religions. Our result was less than studies reported by *Krishnaiah et al. (2004)* nine from India 1,555 (99.6%), and *Ronanki et al. (2014)* 11 from South India 353 (99.4%), while it was close to a study reported by *Ali et al. (2013)* 10 from Pakistan 77 (48.7%). The reason behind these disparate results is the significant variance in religion and culture between different countries.

As for attitudes, our study showed that 696 (66%) of students had negative attitudes. In addition, 583 (55%) refused to donate the eyes of their family members (first-degree relatives) after death. In comparison, only 86 (21.1%) and 90 (22.1%) of medical students in Madinah city indicated a willingness to donate their eyes and their relatives' eyes, respectively (*Hameed & Jadidy, 2015*). In the occupied Palestinian territory, a study of university students revealed that 407 (69%) had no interest in donating their corneas (*Al-Labadi et al., 2018*).

However, it was noted that male participants who had heard about eye donation and did not believe they had sufficient information were significantly more likely to have a good knowledge level ($p \leq 0.05$). In addition, participants who obtained information about eye

**Table 7 Correlation between knowledge levels, demographic characteristics and the sources of the information.**

| Variable | Knowledge level, n (%) | | | p-value |
|---|---|---|---|---|
| | Poor | Fair | Good | |
| **Gender** | | | | |
| Male | 767 (77.8) | 38 (52.8) | 2 (100) | |
| Female | 219 (22.2) | 34 (47.2) | 0 (0.0) | **<0.001** |
| **Nationality** | | | | |
| Saudi | 970 (98.4) | 72 (100) | 2 (100) | |
| None-Saudi | 16 (1.6) | 0 (0.0) | 0 (0.0) | 0.544 |
| **Faculty** | | | | |
| Medicine | 405 (41.1) | 40 (55.6) | 1 (50) | |
| Dentistry | 209 (21.2) | 8 (11.1) | 1 (50) | |
| Pharmacy | 173 (17.5) | 10 (13.9) | 0 (0.0) | |
| Nursing | 121 (12.3) | 9 (12.5) | 0 (0.0) | |
| Applied medical sciences | 31 (3.1) | 1 (1.4) | 0 (0.0) | 0.478 |
| Medical rehabilitation (PT, OT, and RT) | 47 (4.8) | 4 (5.6) | 0 (0.0) | |
| **Academic year** | | | | |
| Second | 307 (31.1) | 22 (30.6) | 1 (50) | 0.439 |
| Third | 186 (18.9) | 11 (15.3) | 0 (0.0) | |
| Fourth | 140 (14.2) | 8 (11.1) | 0 (0.0) | |
| Fifth | 129 (13.1) | 6 (8.3) | 0 (0.0) | |
| Sixth | 224 (22.7) | 25 (34.7) | 1 (50) | |
| **Have you heard about eye donation?** | | | | |
| Yes | 309 (31.3) | 60 (83.3) | 2 (100) | |
| No | 677 (68.7) | 12 (16.7) | 0 (0.0) | **<0.001** |
| **Do you think you have enough information regarding eye donation?** | | | | |
| Yes | 23 (2.3) | 10 (13.9) | 0 (0.0) | |
| No | 963 (97.7) | 62 (86.1) | 2 (100) | **<0.001** |
| **What are the sources from which you obtained the information?** | | | | |
| Health workers | 219 (22.2) | 39 (54.2) | 1 (50) | 0.82 |
| Friends/relatives | 172 (17.4) | 18 (25) | 2 (100) | **0.003** |
| Social media | 270 (27.4) | 32 (44.4) | 2 (100) | **0.001** |
| Television | 84 (8.5) | 12 (16.7) | 0 (0.0) | **0.061** |
| Eye donation campaign | 68 (6.9) | 12 (16.7) | 0 (0.0) | **0.009** |
| Posters | 71 (7.2) | 12 (16.7) | 1 (50) | **0.001** |

Note:
Physiotherapy (PT). Occupational therapy (OT). Respiratory therapy (RT). *Kruskal-Wallis test.

donation from friends/relatives or social media also had a significantly higher probability of having a good knowledge level ($p \leq 0.05$). Second-year male medical students with Saudi nationality who did not obtain information from the health worker were significantly more likely to have a positive attitude ($p \leq 0.05$). This statistical significance may not necessarily reflect a realistic indicator due to the unequal distribution of the study sample, as the

**Table 8 Correlation between attitudes, demographic characteristics and the sources of the information.**

| Variable | Attitude level, n (%) | | p-value |
|---|---|---|---|
| | Negative | Positive | |
| **Gender** | | | |
| Male | 510 (73.3) | 297 (81.6) | |
| Female | 186 (26.7) | 67 (18.4) | **0.003** |
| **Nationality** | | | |
| Saudi | 690 (99.1) | 354 (97.3) | |
| Non-Saudi | 6 (0.9) | 10 (2.7) | **0.017** |
| **Faculty** | | | |
| Medicine | 278 (39.9) | 168 (46.2) | |
| Dentistry | 157 (22.6) | 61 (16.8) | |
| Pharmacy | 100 (14.4) | 83 (22.8) | |
| Nursing | 107 (15.4) | 23 (6.3) | |
| Applied medical sciences | 30 (4.3) | 2 (0.5) | **<0.001** |
| Medical rehabilitation (PT, OT, and RT) | 24 (3.4) | 27 (7.4) | |
| **Academic year** | | | |
| Second | 192 (27.6) | 138 (37.9) | **<0.001** |
| Third | 126 (18.1) | 71 (19.5) | |
| Fourth | 113 (16.2) | 35 (9.6) | |
| Fifth | 100 (14.4) | 35 (9.6) | |
| Sixth | 165 (23.7) | 85 (23.4) | |
| **Have you heard about eye donation?** | | | |
| Yes | 243 (34.9) | 128 (35.2) | |
| No | 453 (65.1) | 236 (64.8) | 0.935 |
| **Do you think you have enough information regarding eye donation?** | | | |
| Yes | 22 (3.2) | 11 (3) | |
| No | 674 (96.8) | 353 (97) | 0.902 |
| **What are the sources from which you obtained the information?** | | | |
| Health workers | 188 (72.6) | 71 (19.5) | **0.007** |
| Friends/relatives | 133 (19.1) | 59 (16.2) | 0.244 |
| Social media | 206 (29.6) | 98 (26.9) | 0.361 |
| Television | 61 (8.8) | 35 (9.6) | 0.647 |
| Eye donation campaign | 55 (7.9) | 25 (6.9) | 0.545 |
| Posters | 55 (7.9) | 29 (8) | 0.97 |

**Note:**
Physiotherapy (PT). Occupational Therapy (OT). Respiratory Therapy (RT). *Mann-Whitney test.

majority, 807 (76%), are male, 1,044 (98.5%) are Saudi, 446 (42.1%) are medical students, and 330 (31.1%) were in their second year.

To meet the increasing local demand for corneal transplantation, awareness programs and eye donation campaigns targeting the community are needed, as reported in our results, where the most preferred source for receiving more information was eye donation

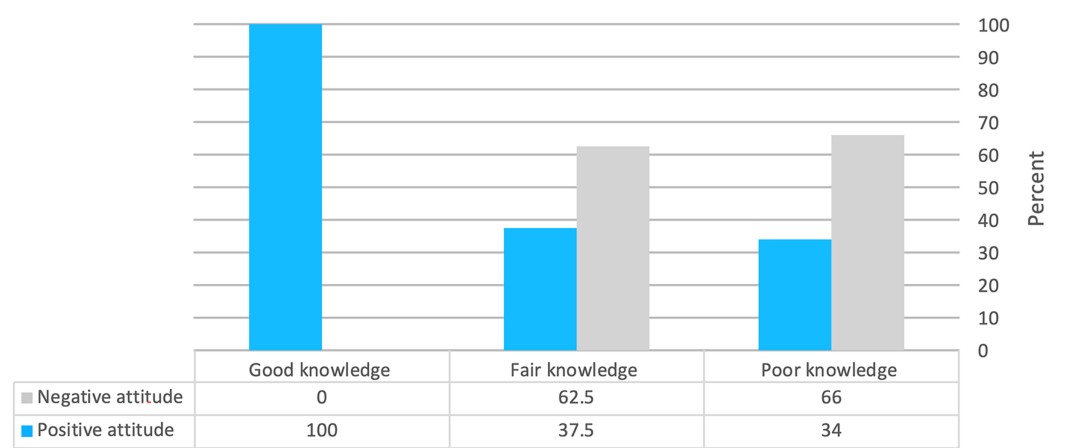

**Figure 2** Correlation between knowledge levels and attitudes.

campaigns 975 (92%). As future healthcare providers, health faculties students must be involved in such programs as they are expected to influence society if they receive a proper education. Most of the studied students, 784 (74%), expressed willingness to educate others about eye donation if they get more information. These multiple well-established findings highlight the importance of spreading awareness to increase the number of local donors.

Establishing effective policies can also help address the organ donor shortage. Many developing countries such as Argentina (*BORA, 2005*), Chile (*LEY-20413 15-ENE-2010 MINISTERIO DE SALUD, SUBSECRETARÍA DE SALUD PÚBLICA—Ley Chile— Biblioteca del Congreso Nacional, 2017*), and Colombia have adopted an "opt-out" policy, whereby all people over the age of 18 are considered organ donors unless they document their refusal. In the developed world, about 24 countries have adopted a presumed consent (opt-out) policy, most notably in Spain, Austria, Belgium, and the United States of America, leading to high donor rates in these countries (*Desde hoy la donación de órganos es obligatoria, 2017*; *Ahmad et al., 2019*).

The present study appeared to have several strengths. The comprehensive study questionnaire addressed almost all items related to eye donation knowledge. Also, an ophthalmologist and a medical research expert have validated the study questionnaire. To the best of our knowledge, this study is the first to study the awareness and attitudes towards eye donation among all health faculty students in Saudi Arabia. Disseminating these findings at the university level is as essential as encouraging the students to get more information regarding eye donation and corneal transplants in Saudi Arabia.

This study faced some limitations. First, we used a cross-sectional design; therefore, the variables were collected simultaneously, so it does not allow for deductions on cause and effect, and the data was obtained by self-report questionnaires, which is prone to recall bias and inaccuracy. Secondly, it is a single-center study, so the student's knowledge may be better in other universities in Saudi Arabia or *vice versa*.

## CONCLUSION

Based on our study findings, students have shown low knowledge and negative attitudes towards eye donation. Therefore, providing students with a basic understanding of eye donations is crucial by adding these concepts to the curriculum, planning awareness-raising events and campaigns, and encouraging their voluntary participation to raise general awareness to attain more local corneal donations.

### Funding

The authors received no funding for this work.

### Competing Interests

The authors declare that they have no competing interests.

### Author Contributions

- Mahmood Showail conceived and designed the experiments, performed the experiments, authored or reviewed drafts of the article, and approved the final draft.
- Turki A. AlAmoudi conceived and designed the experiments, performed the experiments, analyzed the data, prepared figures and/or tables, authored or reviewed drafts of the article, and approved the final draft.
- Esraa Basalem performed the experiments, analyzed the data, prepared figures and/or tables, authored or reviewed drafts of the article, and approved the final draft.
- Khalid Alshebl analyzed the data, prepared figures and/or tables, and approved the final draft.
- Nawaf Meshal Almalki analyzed the data, prepared figures and/or tables, and approved the final draft.
- Abdullah Al Matrafi analyzed the data, prepared figures and/or tables, and approved the final draft.
- Mohammed Ashour analyzed the data, prepared figures and/or tables, and approved the final draft.

### Human Ethics

The following information was supplied relating to ethical approvals (*i.e.*, approving body and any reference numbers):

The research ethics committee of King Abdulaziz University (KAU) in Jeddah, Saudi Arabia granted ethical approval to carry out the study within its facilities (Ethical Application Ref: 14440624).

### Data Availability

The raw data is available in the Supplemental File.

## Supplemental Information

Supplemental information for this article can be found online at http://dx.doi.org/10.7717/peerj.17334#supplemental-information.

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
