# Peer review of "Awareness and attitudes towards eye donation among medical and allied health students in Jeddah, Saudi Arabia"

_PeerJ, doi:10.7717/peerj.17334_

## Round 0.1 · original submission · Major Revisions

Thank you for submitting your manuscript. After careful consideration of feedback from the reviewers, several areas have been identified for improvement. The primary concerns raised include the clarity and detail in the literature review, specific issues in the presentation of data in the abstract, figures, and tables, and the depth of the discussion section. Both reviewers highlighted the need to clarify certain aspects of the experimental design, with particular emphasis on the justification for including specific participant groups and the representation of data. They also suggested updating some references, addressing inconsistencies in the data sheet, refining the conclusion, and discussing study limitations. Addressing these areas will substantially enhance the quality and coherence of your paper.

**Language Note:** PeerJ staff have identified that the English language needs to be improved. When you prepare your next revision, please either (i) have a colleague who is proficient in English and familiar with the subject matter review your manuscript, or (ii) contact a professional editing service to review your manuscript. PeerJ can provide language editing services - you can contact us at copyediting@peerj.com for pricing (be sure to provide your manuscript number and title). – PeerJ Staff

Reviewer 1 ·

Basic reporting

Language is unambiguous, professional.
Literature could say more about study design and rigor of cited studies.
Specific issues include:
Abstract: It would be more helpful to the reader if the percentages were allocated to the concept directly. E.g., line 44 and 45. Best practice is to state 93% (n = XX) had poor knowledge, 6.8% (n = xx) reported fair knowledge etc.
Also, it is usual to round up or down percentages (so 6.8 would be 7) etc. This needs to be addressed across the paper.

The introduction needs to tell the reader about the current situation in SAE. What is the current need? What is the current (or historical) donation rate.

The discussion section is very weak (mainly reporting more findings) and drawing unsubstantiated parallels with research carried out in other countries without providing some context and design comment of the cited knowledge base .

There are other issues that I have highlighted on the PDF with notes

Experimental design

Please confirm that the study design was a cross sectiionl survey.
The paper reports a cross sectional survey.
The study design is replicatable based on the information provided.

However, I do not see a research question and study objectives (these are missing) ?
There is a lack of justification as to why Allied Health Professional students are included. What role would they have in discussing eye donation?
Would you expect students in year 1 to have the same knowledge as those in year 5 or 6?

Validity of the findings

All data is supplied but there are some issues:
Figure and tables have no titles, or legends (there is a file name, but nothing on the tables and figure themselves).
Figure 1, what do the figures e.g., 65.5 etc, refer to?
Figure 2 does not appear to illustrate any statistical results as stated in the narrative line 191) (I am not clear on what Series 1 refers to)?
Figure 3 reports a chi square result.
There are errors in Table 4 (please see highlighted section, this figure is incorrect 283 926.7).
You have not stated what data did not meet parametric requirements.

Results, Line 188 – 190 reports very specific finings from a very specific sub group of your population. These findings are not specifically addressed in the discussion.
The discussion section is weak in so much as it reports further findings or repeats findings. As we do not know much about the context of SAE, making reference to other populations and making comparisons does not expand the knowledge base, it just reports that things are different in other countries (as we might expect as cultures differ).
There is a key unsubstantiated comment (line 209-211). If the authors draw parallels with research carried out in other countries the reader needs to see some context and comment re the rigor of the cited literature

Additional comments

There are other points that I have highlighted on the PDF with notes

Annotated reviews are not available for download in order to protect the identity of reviewers who chose to remain anonymous.

Reviewer 2 ·

Basic reporting

No comments

Experimental design

No comments

Validity of the findings

No comments

Additional comments

A Well executed manuscript with findings and promising action points to mitigate corneal blindness

·

Basic reporting

Introduction:
1. Reference 3 and 4 are old. It will be better to use recent reference.
2. Reference 7 is old. Better reference like https://restoresight.org/members/publications/statistical-report/ can be used.
3. Purpose should be more clear why health care students are involved. There is already one study (reference no.21) done on health care students in Saudi.
Raw Data set
One student mentioned age as zero in data sheet provided. Kindly check the data sheet.

Experimental design

Methodology:
1. Whether pilot study data was included in final study results?
2. Sample size estimation and sampling technique is not mentioned. Need to mention that. That will help reader to know that whether study finding can be generalized or not and bias is there in the study or not.

Validity of the findings

Result:
1. Some repetition of data in table and figure can be avoided in text.
2. One student mentioned age as zero in data sheet provided. Kindly check the data sheet.
3. Based on excel data from medicine 6th year student 117 students were not heard about eye donation is quite surprising. They must have studied ophthalmology till now.
4. Line 184-186. Table 6 shows that male participants, participants who had heard about eye donation, and those who did not think they have enough information regarding eye donation were significantly more likely to have a good knowledge level. The meaning of sentence is not clear how the students who think they had not enough knowledge is having good knowledge. Kindly justify.
5. The number of male students are more than female so obviously students having poor knowledge will be more in males.
6. Table 7- age which category were used to apply chi square test. The data presented is quantitative data (Mean and SD) but test mentioned is chi square test which is a test for qualitative data. Kindly justify.
Discussion:
1. It would have better to compare the knowledge with students preferably with health sciences student. Comparing with general population will not be much useful.
2. It will be better along with comparison, the factors which may be responsible for such differences could be discussed.
3. All result to be discussed preferably. Some of the important significant differences results are not discussed.
4. These students are health sciences students, few from medicine also where ophthalmology is subject. In spite of that why the awareness is such low needs to be discussed.
Conclusion:
1. Need to be written shorter and crisp.
Limitations of the study can be mentioned.

Additional comments

The reason for such low knowledge especially amongst health sciences students to be found out and action to be taken accordingly as they are future health care providers.

---

## Round 0.2 · Minor Revisions

Your manuscript has been reviewed, and we believe it holds promise. However, minor revisions are needed before we can move forward with publication. Specifically:

Methodology Clarification: Please revise your sample size calculation method, ensuring its validity.

Results Presentation: Address the data representation issues in Tables 3 and 4 for clarity.

These revisions are essential for the clarity and impact of your work. We look forward to your updated manuscript.

·

Basic reporting

no comments

Experimental design

Methodology:
It has been mentioned that https://www.calculator.net was used to calculate sample size. But on website the sample size calculator is not available. kindly check again.

No reference for sample size estimation used, also no assumption about % of students having knowledge or otherwise not considered. So sample size estimation is wrong. Need to justify or correct it.
Nothing mentioned about sampling technique. Was it random sample or purposive sample? Sampling technique needs to be mentioned.

Validity of the findings

Results:
Does Table 3 mention data of 371 students who are aware of eye donation (as per table1). Is there any multiple response? Not mentioned in table 3 regarding this. If its multiple response, there is need to mention that in table and also to specify is column heading n=371. In title n=1060 is mentioned if it is so then it’s wrong as out of 1060 only 371 knows about eye donation, so source of information to be asked to them only not to those who do not know about eye donation.
Table 4 mention 643 are willing to donate eyes, but as per table 5 reason for not willing to donate eyes is asked to all including those who are willing to donate. Kindly justify.

Additional comments

Discussion:
These students are health sciences students, few from medicine also where ophthalmology is subject. In spite of that why the awareness is such low needs to be discussed elaborately. Even their curriculum needs to be reviewed as 23 % students already had ophthalmology posting being in 6th academic year.

---

## Round 0.3 · accepted · Accept

Dear Dr. AlAmoudi,

I am writing to inform you that I have taken over the editorial responsibilities for your paper, as the previous editor is unavailable. I have thoroughly reviewed your manuscript independently and have also considered the comments and recommendations from the peer reviewers.

I am pleased to advise that the above paper has now been accepted for publication in PeerJ. Thank you for giving the Journal the opportunity to publish your work.

Best Regards,
Andree Hartanto

·

Basic reporting

no comments

Experimental design

no comments now. necessary modifications done.

Validity of the findings

necessary modifications done.

Additional comments

no.